

# Construction and validation of a risk prediction model for postoperative pulmonary infection in patients with brain tumor: a retrospective study

Jiangling Lan[1], Xing Liu[2], Ligen Mo[1], Dandan Wei[1], Shizhen Zhang[3], Yujiao Zhang[3], Yin Zhu[3] and Yi Lei[1]

[1] Guangxi Medical University Cancer Hospital, Nanning, China
[2] The People's Hospital of Guangxi Zhuang Autonomous Region, Nanning, China
[3] Guangxi Medical University, Nanning, China

## ABSTRACT

**Objectives.** This study aimed to investigate the influencing factors and construct a risk prediction model for postoperative pulmonary infection in patients with brain tumor.

**Methods.** This investigation encompassed a cohort of 636 individuals who were diagnosed with brain tumors and underwent surgical treatment between October 2019 and October 2023. According to the ratio of 7:3, the patients were randomly divided into training set and validation set. Univariate analysis and multivariate Logistic regression analysis were performed on the data in the training set. Finally, the independent risk factors of postoperative pulmonary infection in patients with brain tumor were screened out. R software was used to establish a nomogram model for predicting the risk of postoperative pulmonary infection. Receiver operating characteristic (ROC) curve, calibration curve and Hosmer-Lemeshow test were used to evaluate the discrimination and calibration of the model. Decision curve analysis was used to evaluate the clinical benefit of the model.

**Results.** The prevalence of postoperative pulmonary infection in patients with brain tumors was 17.9%. The nomogram contained several independent risk factors: age $\geq$ 60 years, diabetes mellitus, GCS score < 13 points, postoperative bedtime, and postoperative D-Dimer. The prediction model yielded an area under the curve (AUC) of 0.814 (95% confidence interval CI [0.756–0.873]) in the training set, and an AUC of 0.752 (95% CI [0.653–0.850]) in the validation set. The $P$-values for the Hosmer-Lemeshow test in the training set are 0.629, while in the validation set, they are 0.128. Decision curve analysis demonstrated that the model's clinical effectiveness is satisfactory.

**Conclusions.** Age $\geq$ 60 years, diabetes mellitus, GCS score < 13 points, postoperative bedtime and postoperative D-Dimer are risk factors for postoperative pulmonary infection in patients with brain tumor. The developed prediction model demonstrates substantial predictive value and clinical applicability, serving as a valuable reference for medical professionals in recognizing postoperative pulmonary infections in patients with brain tumors and facilitating preventive nursing measures.

Corresponding author
Yi Lei, leiyi@sr.gxmu.edu.cn

## INTRODUCTION

Recent global cancer statistics indicate that in 2020, there were 308,000 newly diagnosed cases of brain tumors and 251,000 fatalities across the globe (*Sung et al., 2021*). It is increasing year by year, and more than half of them are malignant tumors, with high incidence, high mortality, and high disability rate (*Longo & Agarwal, 2019*). Currently, patients with brain tumors have access to a diverse array of treatment modalities, including surgical intervention, radiotherapy, chemotherapy, biological immunotherapy, and molecular targeted therapy. However, surgical treatment is still the preferred treatment for most patients with brain tumors (*National Health Commission medical Administration, Brain Glioma Professional Committee of Chinese Anti-Cancer Association & Brain Glioma Professional Committee of Chinese Medical Doctor Association, 2022*).

The significance of surgery in enhancing patient survival rates cannot be overstated; however, the prevalence of postoperative complications remains a considerable issue, presenting substantial challenges to the rehabilitation process of patients. Pulmonary infection represents a prevalent postoperative complication (*Zuo et al., 2019*), capable of exacerbating the patient's condition, extending the recovery period, inflating hospitalization costs, and in severe instances, leading to respiratory failure and mortality (*de la Gala et al., 2017*). In comparison with pulmonary infections encountered in general surgery, those occurring in patients with brain tumors within the realm of neurosurgery present a considerably graver scenario. Given the unique characteristics of the disease, individuals diagnosed with brain tumors may experience disturbances in consciousness, accompanied by symptoms such as vomiting, a diminished or absent cough reflex, and swallowing difficulties. These conditions often coincide with fluctuations in intracranial pressure, thereby markedly elevating the risk of pulmonary infections (*Karhade et al., 2017*; *González-Bonet, Tarazona-Santabalbina & Tudela, 2016*). It is reported that the incidence of pulmonary infection after brain tumor operation was as high as 22.86% (*Zhang & Zhang, 2015*). The mortality rate associated with postoperative pulmonary infections has been reported to be between 20% and 50% in patients undergoing surgery (*Russotto, Sabaté & Canet, 2019*). Therefore, early identification of the risk factors related to postoperative pulmonary infection in patients with brain tumor has important clinical value for controlling and preventing the occurrence of pulmonary infection.

The occurrence of postoperative pulmonary infection in individuals with brain tumors is influenced by a multitude of factors. According to *Alkins et al. (2021)*, patients with both brain tumors and diabetes mellitus are more likely to get pulmonary infection. *Longo & Agarwal (2019)* demonstrated a significant correlation between postoperative pulmonary infection and factors such as age, operation time exceeding 3 h, gender, history of chronic lung illness, and body mass index (BMI). A recent meta-analysis indicated that a chronic pulmonary history, diabetes mellitus, cardiovascular disease history, age, operation time,

and Glasgow Coma Scale (GCS) score serve as risk factors for pulmonary infection following brain tumor surgery (*Lan et al., 2023*). The prior research has facilitated clinical personnel in grasping pertinent risk factors to a degree. Nonetheless, the extent to which each risk factor impacts postoperative pulmonary infection in individuals with brain tumors remains ambiguous, rendering stratified management unfeasible. Currently, there exists a lack of pertinent research regarding the predictive model for postoperative pulmonary infections in individuals diagnosed with brain tumors. There is a lack of risk assessment tools to guide medical staff to conduct early individualized assessment on patients with brain tumor surgery.

Hence, the objective of this project is to develop and validate a risk prediction model for forecasting postoperative pulmonary infection in individuals following brain tumor surgery. This will serve as a guide for identifying patients at substantial risk and adopting preventive measures.

## METHODS

### Study participants

This study is a retrospective study, and 636 patients who underwent craniotomy for brain tumors in Guangxi Medical University Cancer Hospital from October 2019 to October 2023 were selected as the study objects. This study has been approved by the Ethics Committee of Guangxi Medical University Cancer Hospital (number: KYB2023174). Informed consent was waived with the approval of the Ethics Committee.

The study included patients who met the following criteria: (1) brain tumors were detected using MRI or CT scans; (2) patients who had undergone their first surgery; (3) patients who were 18 years of age or older; (4) patients for whom complete pathological and clinical data were available. The exclusion criteria encompassed the following: (1) presence of tumor involvement in other areas; (2) occurrence of preoperative pulmonary infection; (3) No chemotherapy was given before or after surgery; (4) occurrence of postoperative infection in other areas, such as cerebral infection, surgical incision infection, and urinary tract infection.

### Diagnostic criteria for postoperative pulmonary infection

Postoperative pulmonary infection refers to the new infection in patients undergoing surgery within 30 days after surgery (*The Fourth Committee of the Hospital Infection Control Branch of the Chinese Preventive Medicine Association, Key Site Infection Prevention and Control Group, 2018*). The outcome indicators observed in this study were new pulmonary infections within 30 days after brain tumor surgery. Diagnostic criteria for pulmonary infection: Chest X-ray or CT showed new or progressive infiltrating shadows, solid shadows, or ground glass shadows, clinical diagnosis can be established by adding two or more of the following three clinical symptoms: (1) Fever, body temperature > 38 °C; (2) Purulent airway secretions; (3) Peripheral white blood cell count $>10\times10^9$/L or $< 4\times10^9$/L (*Shi, 2018*).

## Data collection

To ensure the consistency of the study, the participants in this study were uniformly trained to collect clinical data through the hospital electronic medical record system. Through literature research, expert group discussion and clinical practice, the content of clinical data collection was determined, including (1) Basic information of patients: age, sex, BMI, smoking history, indwelling gastric tube; (2) preoperative combination of basic diseases: diabetes mellitus, pulmonary chronic disease, cardiovascular disease; (3) preoperative laboratory indicators (all were the results of the first examination after admission): preoperative albumin, preoperative prealbumin, preoperative D-Dimer; (4) operative related factors: American Society of Anesthesiologists (ASA) classification, operative time, mechanical ventilation time; (5) postoperative factors (postoperative laboratory indicators were the results of postoperative examination within 3 days): postoperative albumin, postoperative prealbumin, postoperative D-Dimer, postoperative GCS score, and postoperative bedtime. The initial data is inputted and authenticated by two individuals using a standardized coding process. Excel software was utilized to create a database, which was then imported into the R software for additional data analysis. Various techniques were employed to address the issue of missing values based on distinct factors. The median was utilized to impute missing values in continuous data, while the mode was employed for categorical variables.

## Statistical analysis

Statistical analysis was performed using R 4.3.2 software (*R Core Team, 2023*). The measurement data conforming to normal distribution were described by mean ± standard deviation, two independent samples $t$-test was used for comparison between the two groups. Continuous variables that did not conform to the normal distribution were described by the median and quartile spacing, and the rank sum test was used for comparison between groups. Counting data were described by frequency and percentage, and the chi-square test was used for comparison between groups.

The training set and validation set were randomly divided in a 7:3 ratio. The dependent variable in the study was the occurrence of postoperative pulmonary infection in patients in the training set. The independent variables comprised the factors that influenced this occurrence. The variables underwent an initial screening using univariate Logistic regression. Only the variables with a significance level ($P < 0.05$) were included in the subsequent multivariate analysis. The researchers employed multivariate Logistic regression to conduct a secondary screening of variables. Using the stepwise technique with an inclusion threshold of 0.05 and an exclusion threshold of 0.10 (*Wu et al., 2022*), they identified the independent risk factors for postoperative pulmonary infection. Finally, a nomogram of pulmonary infection after brain tumor surgery was constructed based on the results of Logistic regression analysis. The prediction model was evaluated by discrimination, calibration, and clinical practicability. The area under receiver operating characteristic curve (AUC) was used to evaluate the model's discrimination. The AUC range was 0.5~1, and AUC > 0.7 indicated that the model had a good discrimination (*Zhou et al., 2019*). The calibration curve and Hosmer-Lemeshow test were used to

evaluate the calibration of the model. Decision curve analysis was used to evaluate the clinical effectiveness of the model. The MASS package in R software was used for multivariate Logistic regression analysis, and the "rms" package was used to build the nomogram. Receiver operating characteristic (ROC) curve in model validation was drawn with "pROC" package, calibration curve was drawn with "rms" package, and decision curve was drawn with "rmda" package. In this study, bilateral $P < 0.05$ was considered statistically significant.

## RESULTS

### The incidence of postoperative pulmonary infection

760 patients undergoing surgery for brain tumors were enrolled between October 2019 and October 2023. Of these, 124 patients who met the exclusion criteria were removed from the study (93 patients combined with tumors of other sites, 17 patients with pulmonary infection before surgery, and 14 patients with infection at other sites after surgery), and finally 636 patients were found to be eligible for analysis (Fig. 1). Among them, 291 were men and 345 were women, including 232 gliomas, 223 meningiomas, 92 sellar tumors, 66 schwannomas, and 23 other types of tumors. Pulmonary infection occurred in 114 patients (17.9%). Patients were randomly divided into the training set and the verification set according to the ratio of 7:3, and the division of the training set and the verification set was shown in Table S1.

### Clinical data of patients with and without pulmonary infection in training set

When comparing the clinical data of patients with and without pulmonary infection in the training set, the results indicated that there were statistically significant differences in the distribution of 11 variables. These variables included age, GCS score, mechanical ventilation time, indwelling gastric tube, diabetes mellitus, cardiovascular history, preoperative D-Dimer, postoperative albumin, postoperative prealbumin, postoperative D-Dimer, and postoperative bedtime. Patients with pulmonary infection and non-pulmonary infection showed significant differences in these variables ($P < 0.05$). However, the remaining eight variables did not show any statistically significant differences (Table 1).

### Univariate analysis

As shown in Table 2, the occurrence of postoperative pulmonary infection in patients with brain tumor was taken as the dependent variable, and 19 influencing factors were taken as independent variables for univariate analysis. The results showed that age, diabetes mellitus, cardiovascular history, indwelling gastric tube, mechanical ventilation time, GCS score, postoperative albumin, postoperative prealbumin, and postoperative D-Dimer, postoperative bedtime were statistically significant ($P < 0.05$).

### Multivariate logistic regression analysis

The candidate variables that met the requirements in the univariate analysis were included in the multivariate logistic regression analysis. The results showed that age, diabetes

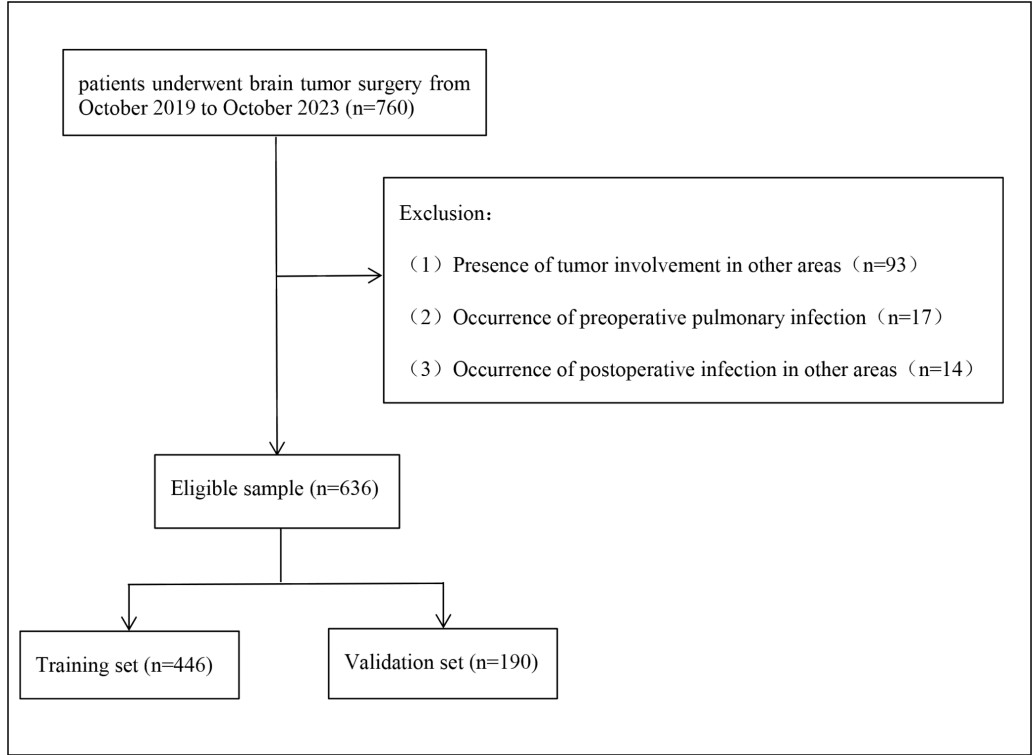

**Figure 1 Flow diagram of the selection of eligible patients.**

mellitus, GCS score, postoperative bedtime, and postoperative D-Dimer were independent risk factors for postoperative pulmonary infection in patients with brain tumors (Table 3).

## A nomogram for postoperative pulmonary infection

Five independent risk factors (age, GCS score, postoperative bedtime, postoperative D-Dimer, diabetes mellitus) selected by multivariate logistic regression analysis were included in the nomogram model as the final predictors. Using R software's 'rms' package to create a nomogram based on the predictors (Fig. 2). The corresponding values of each variable were scored by nomogram, and then the total points were obtained by adding the points of all variables. According to the total points, a vertical line was drawn down to mark the estimated probability of postoperative pulmonary infection in patients with brain tumor.

## Verification of prediction model

The verification of this prediction model is based on the discrimination and calibration of the model. The discrimination of the model is evaluated by drawing the area AUC under the ROC curve for predicting postoperative pulmonary infection in patients with brain tumors. The AUC of the training set and the validation set were 0.814 (95% CI [0.756–0.873]) and 0.752 (95% CI [0.653–0.850]). All of them are greater than 0.7, indicating that the model has a good discrimination (Figs. 3 and 4).

Furthermore, the Hosmer-Lemeshow test demonstrated a satisfactory fit ($P = 0.629$ for the training set; $P = 0.128$ for the validation set), suggesting that there is no notable

**Table 1  Comparison of clinical data of patients with or without pulmonary infection in training set.**

| Variables | No pulmonary infection ($n = 366$) | pulmonary infection ($n = 80$) | $t/z/\chi^2$ | $P$- value |
|---|---|---|---|---|
| Age | | | | |
| ≥60 year | 44(12.02) | 35(43.75) | 45.342 | <0.001 |
| <60 year | 322(87.98) | 45(56.25) | | |
| Sex | | | | |
| Male | 166(45.36) | 36(45.00) | 0.003 | 0.954 |
| Female | 200(54.64) | 44(55.00) | | |
| Smoking history | | | | |
| Yes | 59(16.12) | 10(12.50) | 0.658 | 0.417 |
| No | 307(83.88) | 70(87.50) | | |
| ASA classification | | | | |
| ≥3 | 104(28.42) | 20(25.00) | 0.381 | 0.537 |
| <3 | 262(71.58) | 60(75.00) | | |
| BMI | 23.12 ± 3.60 | 23.44 ± 3.30 | −0.718 | 0.473 |
| Operative time | | | | |
| <3 hour | 32(8.74) | 3(3.75) | 2.263 | 0.132 |
| ≥3 h | 334(91.26) | 77(96.25) | | |
| GCS score | | | | |
| <13 | 9(2.46) | 13(16.25) | 26.626 | <0.001 |
| ≥13 | 357(97.54) | 67(83.75) | | |
| Mechanical ventilation time | 7.00(4.50–10.00) | 8.25(5.00–11.75) | 2.357 | 0.018 |
| Indwelling gastric tube | | | | |
| Yes | 26(7.10) | 19(23.75) | 20.053 | <0.001 |
| No | 340(92.90) | 61(76.25) | | |
| Postoperative bedtime | 4.00(2.00–9.00) | 10.00(5.00–19.00) | 5.581 | <0.001 |
| Diabetes mellitus | | | | |
| Yes | 15(4.10) | 23(28.75) | 51.186 | <0.001 |
| No | 351(95.90) | 57(71.25) | | |
| Pulmonary chronic disease | | | | |
| Yes | 1(0.27) | 1(1.25) | 1.403 | 0.236 |
| No | 365(99.73) | 79(98.75) | | |
| Cardiovascular disease | | | | |
| Yes | 44(12.02) | 23(28.75) | 14.391 | <0.001 |
| No | 322(87.98) | 57(71.25) | | |
| Preoperative albumin | 39.71 ± 3.66 | 39.09 ± 3.11 | 1.409 | 0.160 |
| Preoperative prealbumin | 283.55 ± 86.84 | 273.85 ± 66.89 | 0.940 | 0.348 |
| Preoperative D-Dimer | 0.45(0.18–0.85) | 0.65(0.29–1.71) | 2.654 | 0.008 |
| Postoperative albumin | 33.99 ± 4.07 | 32.57 ± 4.75 | 2.739 | 0.006 |
| Postoperative prealbumin | 210.81 ± 58.35 | 194.32 ± 58.40 | 2.290 | 0.023 |
| Postoperative D-Dimer | 1.64(0.76–2.77) | 2.97(1.42–5.70) | 5.204 | <0.001 |

Notes.

ASA, American Society of Anesthesiologists; BMI, Body Mass Index; GCS, Glasgow Coma Scale.

**Table 2   Univariate analysis of postoperative pulmonary infection in patients with brain tumor.**

| Variables | OR | 95% CI | P- value |
|---|---|---|---|
| Age | 5.692 | 3.309–9.792 | <0.001 |
| Sex | 1.014 | 0.624–1.650 | 0.954 |
| Smoking history | 0.743 | 0.362–1.526 | 0.419 |
| BMI | 1.025 | 0.958–1.097 | 0.472 |
| Diabetes mellitus | 9.442 | 4.651–19.169 | <0.001 |
| Pulmonary chronic disease | 4.620 | 0.286–74.659 | 0.281 |
| Cardiovascular disease | 2.953 | 1.657–5.262 | <0.001 |
| Indwelling gastric tube | 4.073 | 2.124–7.813 | <0.001 |
| Preoperative albumin | 0.954 | 0.892–1.019 | 0.160 |
| Preoperative prealbumin | 0.998 | 0.995–1.002 | 0.345 |
| Preoperative D-Dimer | 1.146 | 0.986–1.333 | 0.076 |
| ASA classification | 0.840 | 0.482–1.462 | 0.537 |
| Operative time | 2.459 | 0.734–8.238 | 0.145 |
| Mechanical ventilation time | 1.018 | 1.006–1.030 | 0.002 |
| GCS score | 7.697 | 3.164–18.724 | <0.001 |
| Postoperative albumin | 0.919 | 0.865–0.977 | 0.007 |
| Postoperative prealbumin | 0.995 | 0.990–0.999 | 0.023 |
| Postoperative D-Dimer | 1.199 | 1.107–1.299 | <0.001 |
| Postoperative bedtime | 1.095 | 1.063–1.129 | <0.001 |

Notes.
ASA, American Society of Anesthesiologists; BMI, Body Mass Index; GCS, Glasgow Coma Scale.

**Table 3   Multivariate analysis of postoperative pulmonary infection in patients with brain tumors.**

| Variables | OR | 95% CI | P- value |
|---|---|---|---|
| Age | 4.525 | 2.387–8.581 | <0.001 |
| Diabetes mellitus | 8.632 | 3.785–19.686 | <0.001 |
| GCS score | 4.255 | 1.457–12.423 | 0.008 |
| Postoperative bedtime | 1.076 | 1.037–1.117 | <0.001 |
| Postoperative D-Dimer | 1.113 | 1.018–1.217 | 0.019 |

Notes.
GCS, Glasgow Coma Scale.

disparity between the projected probability and the actual probability. This indicates that the model exhibits a prominent level of calibration. The calibration curves of both the training set and the validation set closely resemble the ideal curve, indicating a satisfactory level of calibration for the prediction model (Figs. 5 and 6).

In this study, the decision curve analysis was used to evaluate the clinical practicability of the model. The decision curve analysis of the risk nomogram of postoperative pulmonary infection in patients with brain tumor is shown in Figs. 7 and 8. The results show that the net rate of return of the model is higher than that of all patients with postoperative pulmonary infection and no postoperative pulmonary infection, indicating that the model has certain clinical application value and patients can benefit from it.

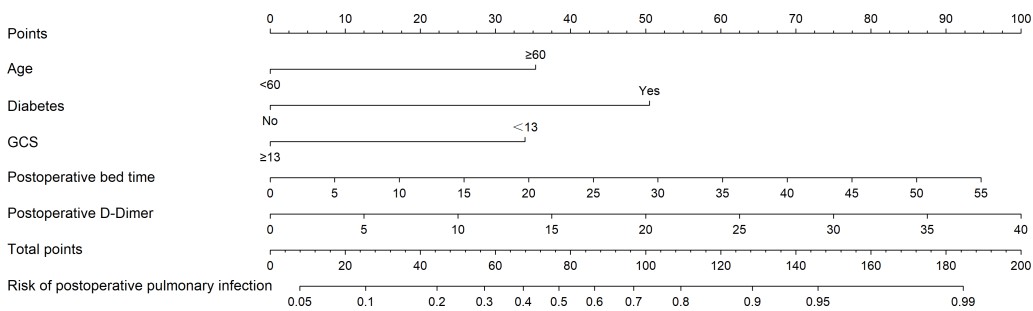

**Figure 2  Nomogram for predicting the risk of postoperative pulmonary infection after brain tumor surgery.** The nomogram was developed in the cohort, with age $\geq$ 60 years, diabetes mellitus, GCS score <13 points, postoperative bedtime, and postoperative D-Dimer. The line segment corresponding to each variable is marked with a scale, which represents the possible value range of the variable. The value of each variable was gave a score on the point scale axis. Total points could be easily calculated by adding each single score and, by projecting the score to the lower total point scale, we were able to estimate the probability of postoperative pulmonary infection in patients with brain tumors.

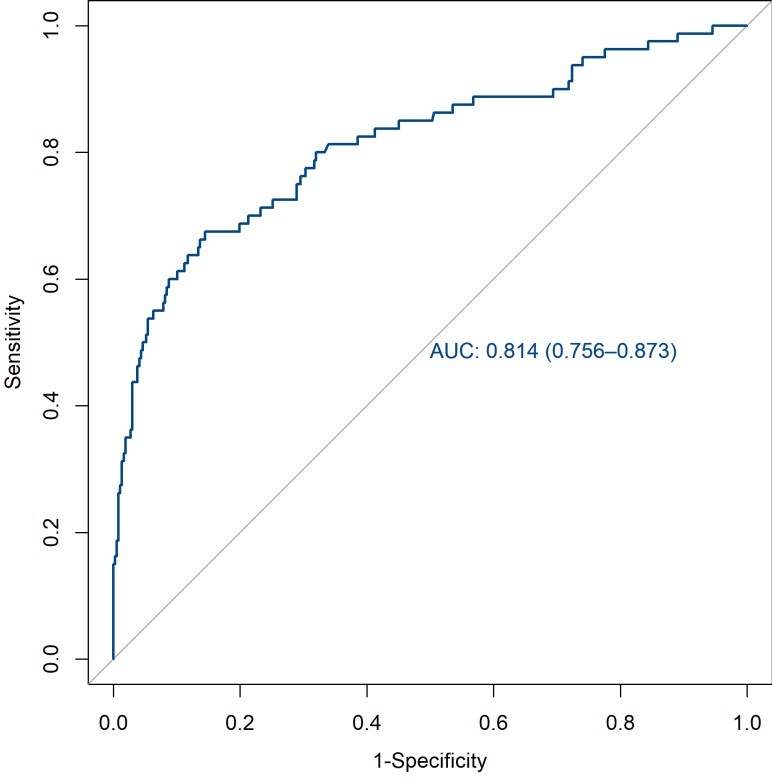

**Figure 3  ROC curve of the training set.** AUC, area under the curve.

## DISCUSSION

This study is the first to construct a risk prediction model for postoperative pulmonary infection in patients with brain tumors. The model contains five predictors: age, diabetes

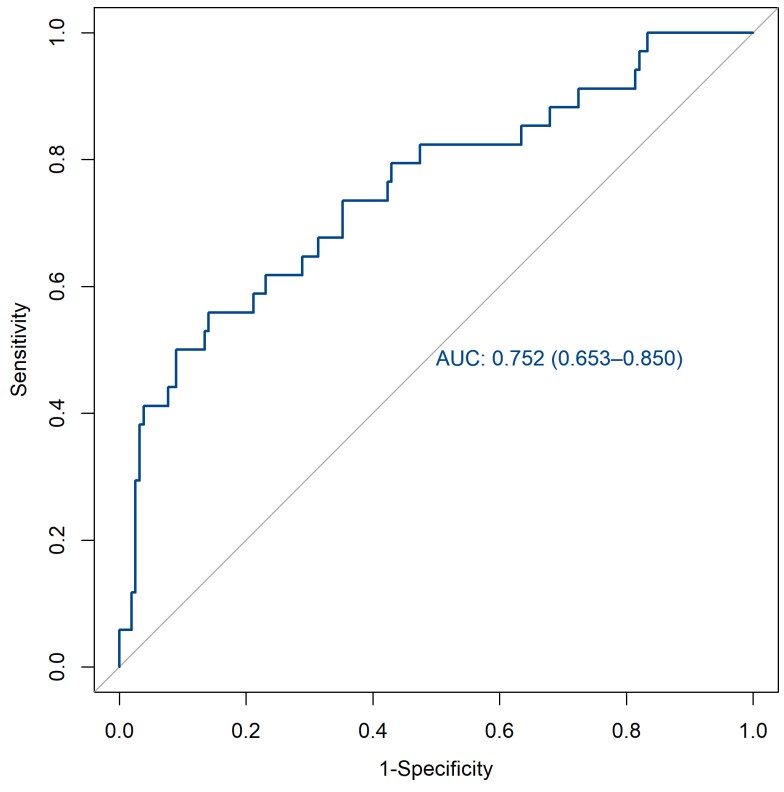

**Figure 4 ROC curve of the validation set.** AUC, area under the curve.

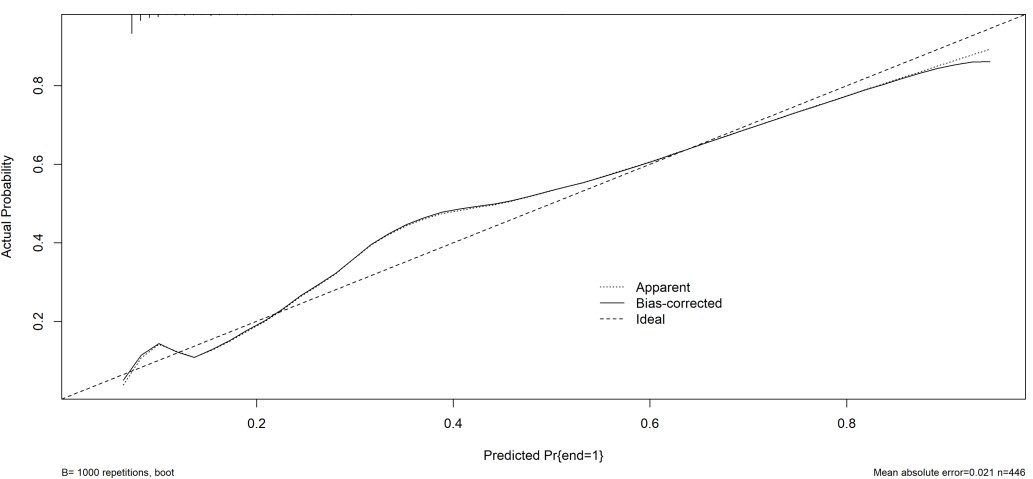

**Figure 5 The calibration plot (training set).** The x-axis represents the predicted the probability of postoperative pulmonary infection in patients with brain tumors. The y-axis represents the actual diagnosed postoperative pulmonary infection in patients with brain tumors. The diagonal dotted line represents a perfect prediction by an ideal model. The solid line represents the performance of the nomogram, of which a closer fit to the diagonal dotted line represents a better prediction.

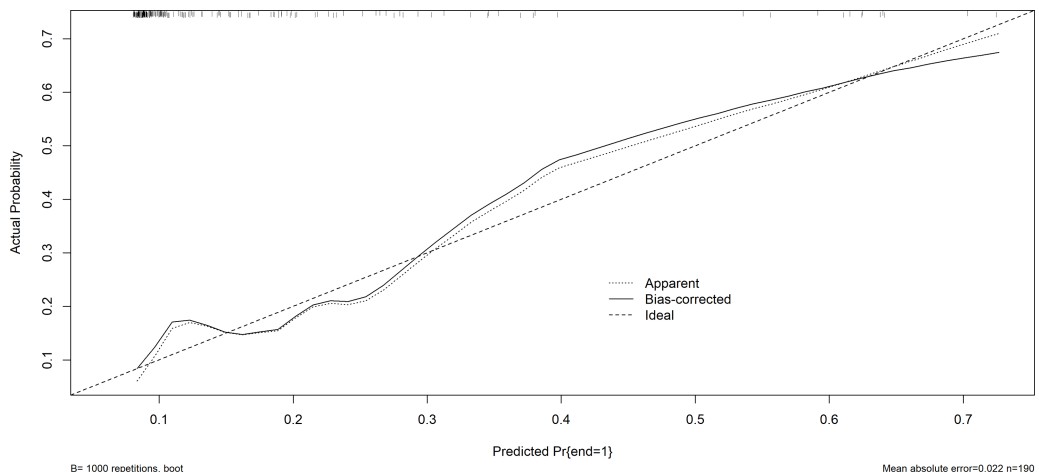

**Figure 6  The calibration plot (validation set).** The x-axis represents the predicted the probability of postoperative pulmonary infection in patients with brain tumors. The y-axis represents the actual diagnosed postoperative pulmonary infection in patients with brain tumors. The diagonal dotted line represents a perfect prediction by an ideal model. The solid line represents the performance of the nomogram, of which a closer fit to the diagonal dotted line represents a better prediction.

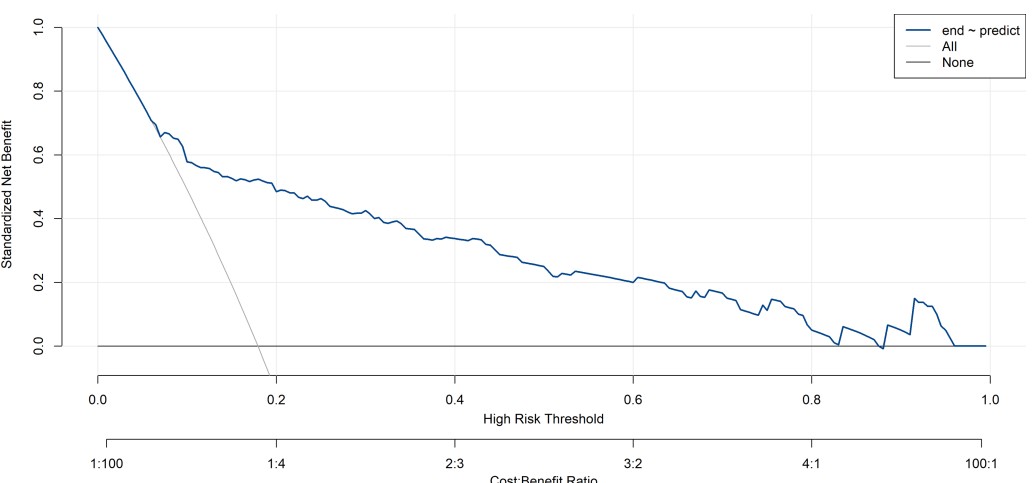

**Figure 7  Decision curve analysis of training set.** The blue line represents the risk of postoperative pulmonary infection after brain tumor surgery nomogram. The thin solid line indicates the assumption that all patients developed postoperative pulmonary infection. The thick line indicates the assumption that no postoperative pulmonary infection occurred in all patients. The area among the model curve "None line" and "ALL line", represents the clinical usefulness of the model.

mellitus, GCS score, postoperative bedtime, postoperative D-Dimer. These data are objective and convenient for clinical collection, which can assess the risk of different patients early and formulate prevention and treatment plans as soon as possible. The nomogram shows good discrimination and calibration and can accurately calculate the

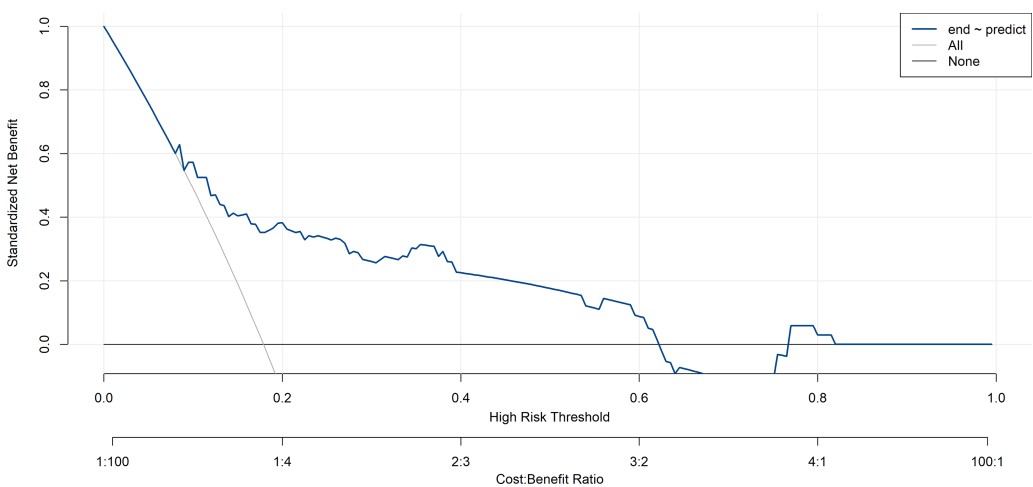

**Figure 8 Decision curve analysis of validation set.** The blue line represents the risk of postoperative pulmonary infection after brain tumor surgery nomogram. The thin solid line indicates the assumption that all patients developed postoperative pulmonary infection. The thick line indicates the assumption that no postoperative pulmonary infection occurred in all patients. The "None" line and "All" line represent the clinical usefulness of the mode.

probability of postoperative pulmonary infection in patients with brain tumor. The decision curve analysis shows that the model has good clinical practical value.

In this study, the incidence of postoperative pulmonary infection was 17.9%, which was like the results reported in previous studies (*Ren et al., 2021*). The risk of postoperative pulmonary infection in individuals with brain tumors is significant, necessitating careful consideration to develop a risk prediction model relevant to this domain in order to mitigate the incidence of pulmonary infections. Currently, the predictive models for pulmonary infections, including deep learning methodologies, present operational challenges and pose difficulties in rendering the prediction outcomes comprehensible to those lacking professional expertise (*Frondelius et al., 2022*). The pulmonary infection risk prediction model developed in this study is straightforward and easily comprehensible, enabling clinical medical personnel to articulate to patients more effectively the methods of preventing pulmonary infection. Furthermore, it allows for a more precise assessment of the risk of pulmonary infection based on patients' clinical data, rather than depending exclusively on prior treatment experiences (*Wang, 2023*). It provides a visual tool for predicting the risk of pulmonary infection after brain tumor surgery and identifying high-risk groups.

Investigating the impact of age on the occurrence of postoperative pulmonary infection in individuals having brain tumor surgery has consistently been a topic of interest. *Wang et al. (2017)* discovered that individuals aged 60 years or older had a 2.85 times greater likelihood of developing postoperative pulmonary infection compared to patients younger than 60 years. *Longo & Agarwal (2019)* examined a total of 28,700 patients who were receiving surgery for brain tumors at 680 medical sites across the United States. The findings indicated that individuals over the age of 60 had a greater likelihood of developing
postoperative pulmonary infection compared to those in other age groups. As individuals age, the respiratory system and immune defense function of older people decline, resulting in a reduced ability to resist bacterial invasion, and the risk of postoperative pulmonary infection is increased (*Liu, 2019*). In addition, the older the age, the accompanying physical complications may also increase, some studies suggest that (*Dang, 2021*), elderly patients are often accompanied by a history of chronic pulmonary disease, preoperative respiratory system has been the existence of chronic inflammation, such as chronic bronchitis, chronic obstructive pulmonary disease. lung tissue ventilation and ventilation capacity decreased, poor elasticity, long-term accumulation of respiratory secretions in the lung is not easy to cough induced infection, the existence of their own complications in elderly patients may also be one of the internal factors to increase the risk of postoperative pulmonary infection. According to the current research results, we still need to be vigilant about the substantial risk of postoperative pulmonary infection in elderly patients with brain tumor and pay enough attention to the preoperative lung function and systemic conditions of elderly patients with brain tumor.

Diabetes is closely related to postoperative pulmonary infection, and hyperglycemia increases the risk of postoperative pulmonary infection (*Longo & Agarwal, 2019*). Long-term hyperglycemia in diabetic patients is easy to drive the proliferation of bacteria. In addition, diabetic patients are prone to peripheral vascular disease, insufficient tissue blood supply, and reduced oxygen concentration, which affects their sensitivity to infection. The decrease of granulocyte chemotaxis and adhesion function reduces the phagocytosis and killing effect on pathogenic bacteria (*López-de Andrés et al., 2019*; *Ma et al., 2019*). *Jin et al. (2023)* found that lung cancer patients with diabetes mellitus had a higher risk of postoperative pulmonary infection. According to the study of *López-de Andrés et al. (2019)* about 118,000 patients undergoing surgical treatment in Spain from 2001 to 2015, the incidence of postoperative pulmonary infection in diabetic patients was 21% higher than that in non-diabetic patients. Therefore, it is particularly important to strengthen the preoperative control of blood glucose in patients with brain tumors complicated with diabetes. The levels of fasting blood glucose or random blood glucose are significantly influenced by lifestyle factors such as dietary choices and physical activity. Assessing the degree of blood glucose regulation over a specific time limit poses significant challenges, potentially compromising the accurate appraisal of preoperative risk and prognosis (*Cavero-Redondo et al., 2017*). HbA1c is the "gold standard" for evaluating blood glucose control in patients with diabetes, which can reflect the average blood glucose level in the past 2–3 months (*Venker, 2017*). Studies have shown that elevated preoperative HbA1c is associated with increased perioperative risk in diabetic patients (*Peng et al., 2021*). Consequently, medical personnel must closely observe the patient's blood glucose levels and the preoperative HbA1c. It is advisable to maintain HbA1c levels below 8.0% and blood glucose within the range of 5.6–10.0 mmol/L prior to surgical procedures (*Wu, Gan & Gao, 2024*).

The Glasgow Coma Scale (GCS) assesses the neurological condition of patients by measuring their reaction in terms of eye opening, language, and limb motor function. The findings of our study indicate that patients exhibiting a GCS score lower than 13

are at an increased risk for developing postoperative pulmonary infections. This could be attributed to patients with brain tumors being more susceptible to surgical complications, potentially resulting in cranial nerve damage during the procedure. Such nerve injuries may subsequently impair the mechanical functioning of the respiratory system (*Vingerhoets & Bogousslavsky, 1994*). A reduced GCS score indicates a more profound impairment of consciousness. This results in extended periods of postoperative bed rest, a heightened probability of relaxation in the tongue muscles, and a descent of the tongue base. The processes of swallowing, coughing, and other reflexive actions exhibit a decline in efficacy, resulting in diminished sputum production. This stagnation of sputum creates an optimal environment for bacterial proliferation, thereby increasing the susceptibility to hypostatic pneumonia. Concurrently, the regurgitation of gastrointestinal contents is susceptible to aspiration, thereby elevating the likelihood of pulmonary infection (*Cardozo Júnior & Silva, 2014*; *Chen et al., 2021*). *Deng, Wang & Zhang (2020)* showed that low postoperative GCS score was an independent risk factor for postoperative pulmonary infection in patients with meningioma. *Dai et al. (2021)* also confirmed that GCS score is closely related to the occurrence of pulmonary infection in patients after craniotomy. For brain tumor postoperative patients with consciousness disorders, respiratory secretions should be cleaned up in time, and patients or their families should be instructed to choose the appropriate eating position (bed elevation 30°–45°), hang a "prevention of aspiration" sign at the bed, and give tube feeding diet combined with parenteral nutrition if necessary to prevent infection induced by aspiration (*Li, 2018*).

*Cao & Ding (2020)* found that bedridden time $\geq 10$ days is a risk factor for secondary pulmonary infection in patients with long-term bedridden stroke, which is like the results of our study. Long-term bedridden patients have an insignificant change in their autonomous position, less thoracic movement, significantly reduced vital capacity and effective ventilation, promoted blood deposition at the bottom of the lung, and the gravity concentration of respiratory secretions, which brought favorable conditions for bacterial reproduction (*Yang et al., 2017*). In addition, the long-term bedridden patients' ability to cough independently is weakened, and they cannot remove viscous secretions. Invasive operations such as sputum suction and indwelling gastric tube are more likely to impose a burden on the normal barrier function of the respiratory tract, leading to bacteria entering the lower respiratory tract and inducing pulmonary infection (*Zheng et al., 2019*). Getting out of bed early can promote the recovery of respiratory function, enhance the contraction ability of patients' respiratory muscles, and discharge secretions easily, which is conducive to preventing pulmonary infection (*Cao et al., 2021*). Medical staff should encourage patients to get out of bed early if the condition permits, to reduce bedtime.

Prior research has demonstrated a positive correlation between elevated postoperative D-Dimer levels and the severity of lung infection (*Huang et al., 2021*). D-Dimer is a byproduct of fibrin that occurs after it is activated and broken down. It is associated with the fibrinolytic system and inflammatory response (*Xiao et al., 2021*; *Yang et al., 2021*). Our investigation discovered that D-Dimer is a distinct risk factor for postoperative pulmonary infection in patients with brain tumors. When the body is invaded by pathogens, pulmonary infection or inflammation occurs, which activates the coagulation and fibrinolytic system

of the body, microthrombus is formed in the pulmonary micro vessels, fibrinogen decomposition, D-Dimer concentration increases, and the formation of pulmonary microthrombus leads to pulmonary arteriolar blockage and affects pulmonary oxygenation function (*Cerda-Mancillas et al., 2020*; *Zhao, Hu & Guo, 2018*). *Zhao et al. (2020)* found that D-Dimer > 2.26 mg/L was an independent risk factor for postoperative pulmonary infection in elderly patients with intertrochanteric fracture. The study of *Ge et al. (2019)* pointed out that the increase of D-Dimer in patients was closely related to inflammation, ICU occupancy and mortality. At present, although D-Dimer alone has a certain accuracy in the diagnosis of pulmonary infection, it is still controversial. It is recommended to combine other indicators to improve the test efficiency (*Li et al., 2022*; *Zheng et al., 2021*). Therefore, we should cooperate to observe the changes of multiple inflammatory indicators in patients and take targeted intervention measures to prevent the occurrence of pulmonary infection in patients with abnormal test results.

## LIMITATIONS

There are still several constraints in this study. Firstly, this study is limited to a single center and does not gather additional external data for external validation. Furthermore, this study was conducted retrospectively, and the data was obtained solely from the electronic medical record system, which posed limitations. Several influential parameters, such as C-reactive protein and procalcitonin in laboratory indicators, were not considered during the analysis, potentially introducing some bias. Furthermore, the research encompassed a diverse array of brain tumor types, and the specific characteristics and dimensions of these tumors were excluded from the analysis, potentially introducing certain limitations. In future research, we shall augment the sample size, conduct prospective multi-center studies, refine the development of a more accurate and clinically pertinent risk prediction model for postoperative pulmonary infection, and devise tailored intervention programs to effectively mitigate the risk of postoperative pulmonary infection in patients with brain tumors.

## CONCLUSION

In summary, age $\geq$ 60 years, diabetes mellitus, GCS score < 13 points, postoperative bedtime, and postoperative D-Dimer were found to be independent risk factors for postoperative pulmonary infection in patients with brain tumor. In addition, the nomogram of postoperative pulmonary infection in patients with brain tumor established in this study is simple and easy to perform, and the prediction effect is good, which can provide reference for identifying high-risk patients and implementing intervention.

### Funding

This study was supported by the Guangxi Medical and Health Appropriate Technology Development and application project (No: S2020104). The funders had no role in study design, data collection and analysis, decision to publish, or preparation of the manuscript.

### Grant Disclosures

The following grant information was disclosed by the authors:
Guangxi Medical and Health Appropriate Technology Development and application project: No: S2020104.

### Competing Interests

The authors declare there are no competing interests.

### Author Contributions

- Jiangling Lan conceived and designed the experiments, performed the experiments, analyzed the data, prepared figures and/or tables, and approved the final draft.
- Xing Liu conceived and designed the experiments, performed the experiments, analyzed the data, prepared figures and/or tables, and approved the final draft.
- Ligen Mo performed the experiments, prepared figures and/or tables, and approved the final draft.
- Dandan Wei performed the experiments, prepared figures and/or tables, and approved the final draft.
- Shizhen Zhang performed the experiments, authored or reviewed drafts of the article, and approved the final draft.
- Yujiao Zhang performed the experiments, authored or reviewed drafts of the article, and approved the final draft.
- Yin Zhu performed the experiments, authored or reviewed drafts of the article, and approved the final draft.
- Yi Lei conceived and designed the experiments, performed the experiments, analyzed the data, prepared figures and/or tables, and approved the final draft.

### Ethics

The following information was supplied relating to ethical approvals (i.e., approving body and any reference numbers):

This study has been approved by the Ethics Committee of Guangxi Medical University Cancer Hospital (number: KYB2023174), informed consent of study subjects was waived due to the retrospective nature of the study.

### Data Availability

The data is available in the Supplementary File.

## Supplemental Information

Supplemental information for this article can be found online at http://dx.doi.org/10.7717/peerj.18996#supplemental-information.

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
