# Peer review of "Construction and validation of a risk prediction model for postoperative pulmonary infection in patients with brain tumor: a retrospective study"

_PeerJ, doi:10.7717/peerj.18996_

## Round 0.1 · original submission · Minor Revisions

The manuscript would benefit from a stronger contextualization of why pulmonary complications are particularly critical in brain tumor surgeries, and how this work advances beyond existing literature in this field.

The methodology section needs greater detail regarding inclusion/exclusion criteria, particularly concerning chemotherapy status and tumor characteristics. While the statistical analysis is robust, the presentation could be enhanced by standardizing references, adding missing figure legends, and improving English language clarity throughout.

The discussion section, while comprehensive, requires condensation to maintain focus on key findings and their clinical implications. Additionally, please ensure all technical terms are consistently defined and citations properly formatted according to journal guidelines.

Reviewer 1 ·

Basic reporting

This study aimed to develop and validate a risk prediction model for postoperative pulmonary infections in patients undergoing brain tumor surgery. The research involved a retrospective cohort of 636 patients who underwent brain tumor surgery and utilized statistical analysis to identify independent risk factors contributing to postoperative pulmonary infections, such as age over 60, diabetes, a Glasgow Coma Scale (GCS) score below 13, extended postoperative bed rest, and elevated postoperative D-Dimer levels. These factors were integrated into a nomogram model, which demonstrated good predictive accuracy and clinical utility through AUC, calibration curves, and decision curve analyses.
Review Report Suggestions:
1. It would strengthen the study to discuss why pulmonary complications are particularly critical in brain tumor surgeries compared to other surgical contexts.
2. The inclusion and exclusion criteria require greater comprehensiveness. Have all factors influencing susceptibility to infection, such as preoperative and postoperative chemotherapy (neoadjuvant and adjuvant chemotherapy), been accounted for either as exclusion parameters or as potential predictors?
3. It is unclear whether the type and size of the tumor were considered in data collection, factors which could influence postoperative outcomes. If not included, please provide a rationale for their exclusion to explain any limitations in risk prediction accuracy.
4. It would be beneficial to discuss the reasoning behind prioritizing this model over other new methods, such as deep learning techniques, which have shown promise in similar predictive modeling fields.
These points could further refine the research presentation.

Experimental design

2. The inclusion and exclusion criteria require greater comprehensiveness. Have all factors influencing susceptibility to infection, such as preoperative and postoperative chemotherapy (neoadjuvant and adjuvant chemotherapy), been accounted for either as exclusion parameters or as potential predictors?

Validity of the findings

3. It is unclear whether the type and size of the tumor were considered in data collection, factors which could influence postoperative outcomes. If not included, please provide a rationale for their exclusion to explain any limitations in risk prediction accuracy.

Additional comments

4. It would be beneficial to discuss the reasoning behind prioritizing this model over other new methods, such as deep learning techniques, which have shown promise in similar predictive modeling fields.

Reviewer 2 ·

Basic reporting

For the most part, the English used in this paper was clear. Some sentences are unclear and the manuscript would benefit from additional checking from an editor fluent in English (e.g., line 65 "...infection will bring burden to the cardiovascular system..."). In line 58, check that this is the right way to cite the "Administration et al. 2022" reference. In general, several references appear to be cited incorrectly (e.g., line 76, "Longo et al." should be "Longo and Agarwal (2021) demonstrated..."). Please check that all references conform to a consistent citation style. The introduction is well-written and clearly defines the problem that this study is trying to address (predicting risk of pulmonary infection following brain tumor resection). It provides enough background to understand the results and their context. I do not see figure legends uploaded. Please provide these. In line 140-141, states what these values represent (i.e., p = 0.05). For the most part, the article is appropriately structured and self-contained. However, the following paper is relevant and overlapping with the contents of this paper, and should be cited: https://doi.org/10.1089/sur.2023.130. It should also be clarified in the introduction how this new study differs and adds novel insight into post-operative pulmonary infections, given that many of the risk factors were previously reported.

Experimental design

As best I can ascertain, this is original primary research within the scope of the journal. The research question (determining the primary factors associated with pulmonary infection post-op and constructing a model that can be used by healthcare professionals to estimate risk) is well-defined, relevant, and meaningful. The methods are clearly described.

Validity of the findings

The raw data is provided in the supplemental materials. Was smoking history factored into this model or the exclusion criteria? If not, why not? This needs to be clarified. The conclusions are appropriately stated in relation to the research question and results.

Additional comments

The discussion section is extremely long, especially in comparison to the length of the results section, and should be severely reduced. For instance, the lengthy discussion of sputum sample collection can likely be removed (lines 257-267).

In lines 293-300, the GCS abbreviation is defined multiple times.

---

## Round 0.2 · accepted · Accept

This manuscript has been improved after revisions. I think this paper can be accepted for publication.

While in production, the authors can add details to figure legends as suggested by Reviewer 2.

Reviewer 1 ·

Basic reporting

Concerns are addressed.

Experimental design

Concerns are addressed.

Validity of the findings

Concerns are addressed.

Additional comments

Concerns are addressed.

Reviewer 2 ·

Basic reporting

Overall, this is an improved revision to the original manuscript. The addition of text to the introduction clarifies the relevance and novelty of this study in relation to prior similar studies on post-surgical pulmonary infection. Additionally, the removal of some of the unnecessary text in the discussion helps highlight the main focus of the manuscript. One minor detail: the figure "legends" that have now been included in this revision are only titles and are not very detailed. Please add detail to describe what each part of the figure represents and any relevant parameters.

Experimental design

The authors have sufficiently addressed the concerns related to experimental design that I raised in my prior comments.

Validity of the findings

The authors have sufficiently addressed the concerns related to the validity of the findings that I raised in my prior comments.

Additional comments

I have no further comments.